# Generation and Prediction of Construction and Demolition Waste Using Exponential Smoothing Method: A Case Study of Shandong Province, China

**Liang Qiao [1,2], Doudou Liu [3], Xueliang Yuan [1,*], Qingsong Wang [1] and Qiao Ma [1]**

[1]  National Engineering Laboratory for Reducing Emissions from Coal Combustion, Engineering Research Center of Environmental Thermal Technology of Ministry of Education, Shandong Key Laboratory of Energy Carbon Reduction and Resource Utilization, Research Center for Sustainable Development, School of Energy and Power Engineering, Shandong University, Jinan 250061, China; qiaol@mail.sdu.edu.cn (L.Q.); wqs@sdu.edu.cn (Q.W.); maq@sdu.edu.cn (Q.M.)

[2]  Shandong Construction and Development Research Institute, Jinan 250004, China

[3]  School of Management Science and Engineering, Shandong University of Finance and Economics, Jinan 250014, China; liudoudou@sdufe.edu.cn

*  Correspondence: yuanxl@sdu.edu.cn; Tel./Fax: +86-531-88395877

**Abstract:** The output of construction and demolition (C&D) waste in China has been rapidly increasing in the past decades. The direct landfill of such construction and demolition waste without any treatment accounts for about 98%. Therefore, recycling and utilizing this waste is necessary. The prediction of the output of such waste is the basis for waste disposal and resource utilization. This study takes Shandong Province as a case, the current output of C&D waste is analyzed by building area estimation method, and the output of C&D waste in the next few years is also predicted by Mann–Kendall trend test and quadratic exponential smoothing prediction method. Results indicate that the annual productions of C&D waste in Shandong Province demonstrates a significant growth trend with average annual growth of 11.38%. The growth rates of each city differ a lot. The better the city's economic development, the higher the level of urbanization, the more C&D waste generated. The prediction results suggest that the output of C&D waste in Shandong Province will grow at an average rate of 3.07% in the next few years. By 2025, the amount of C&D waste will reach 141 million tons. These findings can provide basic data support and reference for the management and utilization of C&D waste.

**Keywords:** construction and demolition waste; trend test; exponential smoothing method; prediction

## 1. Introduction

Construction and demolition (C&D) waste is produced during the process of building construction, expansion, and demolition. Owing to the gradual progress of urbanization construction, the areas of buildings that have been completed and are still under construction have reached 4.1 and 14.1 billion $m^2$, respectively, in 2018. As a result, the production of C&D waste rapidly increases [1]. The estimated annual production of C&D waste in China is approximately 2 billion tons, which accounts for 80–90% of the total municipal waste. However, the rate of resource utilization is less than 5% [2]. Demolition waste mainly consist of concretes, bricks, metals, timbers, plastics, gravels, ceramics, and glasses [3,4]. Most of the compositions of such waste are reusable materials that are usually disposed in landfills and dumps, thereby causing serious environmental and land occupation issues [5]. The disposal and utilization of C&D waste are common concerns of society. The production of C&D waste is the basis for formulating countermeasures for the corresponding treatments and resource utilization. However, because most

provinces and cities in China have no detailed statistical data for the production of C&D waste, no uniform standard for the calculation of the production has been established. The government also lacks information about the production of C&D waste, which increases the difficulty of implementing comprehensive management for these waste [6].

Quantitative waste prediction is crucial for waste management. Apart from estimation and prediction techniques, no method can be used to accurately and easily estimate the amount of waste produced by C&D projects. Estimation involves calculating the historical quantity of building waste and prediction determines the future production of construction waste on the basis of historical data. Three main methods can be used to estimate the output of C&D waste, namely, building area estimation method, material flow analysis (MFA) approach, and geographic information system (GIS) method [7–9].

The building area estimation method considers the various stages of C&D waste generation from the perspectives of construction, demolition, and decoration and has become the mainstream method for estimating construction waste. Forecasting C&D waste generation can be performed using gray models, linear regression, and autoregressive integrated moving average model. Zhu [10] used the building area method to estimate the production of C&D waste generated during the construction, demolition, and decoration stages of Guangzhou. On the basis of the analysis of the effect of C&D waste production, a regression model was used to predict the future output of these waste. Chen [11] and Wang [12] estimated and analyzed the output of C&D waste in Chengdu and Xi'an, respectively, using the building area method and predicted the output using the GM (1, 1) gray model. P.V. Sáez et al. [13] conducted a weight-per-construction-area method by considering several parameters and quantifying the waste generated by the construction activities in the Mediterranean using the total building area and dwellings number of the projects. Wang [6] applied the gray Verhulst prediction model to forecast the C&D waste output in Shenyang. The results showed that the gray Verhulst prediction model has higher accuracy than the GM (1, 1) method and reasonably reflects the change trend of the C&D waste output. Zhou et al. [14] used the building area estimation method to estimate the annual output of C&D waste in China and subsequently applied the GM (1, 1) model for forecasting and analysis. Zheng et al. [7] conducted an explicit analysis on the basis of a weight-per-construction-area method and reported that between 2003 and 2013, China generated about 2.36 billion tons of C&D waste every year; demolition and construction waste accounted for 97% and 3% of the total waste, respectively, in 2013.

MFA is a suitable method for modeling waste management systems because it supports waste management decision-making from the viewpoint of material recycling [8]. Cochran and Townsend [9] evaluated the performance of an MFA approach in estimating the C&D debris generation and composition in the United States. Historical consumption data and average life estimates of construction materials are used to estimate the amount of debris produced by demolition activities.

GIS is a computer-based tool that is used to collect, store, integrate, process, and analyze geospatial information data and express data in space [15]. This technology has been widely used in municipal solid waste management [16–18]. Chalkias and Lasaridi [19] identified two main classifications of GIS-based waste management applications. The first is GIS for selecting the location of waste facilities, and the second is waste management application supported by GIS related to waste collection [20]. The latter includes proposals from developing and high-income countries [21–25]. Gallardo et al. [26] suggested that the waste spatial distribution in a specific geographic area can be obtained by analyzing the corresponding waste output, composition, and change all the year using GIS. Wu et al. [27] proposed a new approach to quantify the demolition waste from generation to final disposal. This method first defines the research issues to establish the objectives and subsequently imports the DW-GIS database into the GIS software.

Exponential smoothing method is a commonly used prediction method based on time series. Taylor [28,29] adapted the Holt–Winters exponential smoothing formulation to forecast short-term electricity demand, and applied total and split exponential smoothing method to forecast the monthly sales data of a publishing company. Guan et al. [30] studies showed that the exponential smoothing method based on the additive model method can be effectively applied to the time series analysis of

brucellosis in China. The quadratic exponential smoothing method is applied to medium and short term forecast of some industries. Raha and Gayen [31] used the exponential model to simulate the meteorological drought in Bankura District at several time steps. Qian [32] forecasted the trend of regional difference of rural residents' income using the quadratic exponential smoothing method. Qin et al. [33] used the quadratic smoothing index method and Kalman filter model method to predict the short-term traffic volume of the road cross section.

Many studies utilized the estimation methods and prediction models of C&D waste generation. Given that the consumption of construction materials is greatly affected by the change in building structure [34], the GIS-based prediction model requires a large number of comprehensive basic data that are difficult to obtain. On the contrary, the data required by the area estimation method are often provided in national and provincial statistical yearbooks, which are relatively comprehensive and accurate [3,10]. Studies involving the prediction method for C&D waste mainly focus on mid- and long-term predictions but lack information on the short-term aspect. Therefore, it is very important to use the open data of the statistics department to estimate the output of C&D waste accurately and efficiently, and to find a prediction model to predict the output of C&D waste feasibly, which is helpful to promote the resource utilization of C&D waste. The purpose of this paper is to establish a methodology to estimate and predict the output of C&D waste. Firstly, according to the annual construction area and the demolition area, the building area method is used to estimate the amount of C&D waste in Shandong Province and the 17 cities from 2000 to 2017. Then, Mann–Kendall trend test method was used to test the change trend of C&D waste output. Finally, in view of the time series characteristics of linear trend of the estimated C&D waste output, the quadratic exponential smoothing method is applied. Based on the analysis of estimation and prediction results, some suggestions are put forward to promote the treatment and resource utilization of C&D waste.

## 2. Methods

In order to predict the production of C&D waste, this study first estimated the output of C&D waste at present by building area estimation method, and then established a prediction model based on the quadratic exponential smoothing method through the analysis of the trend of the output of C&D waste (Figure 1).

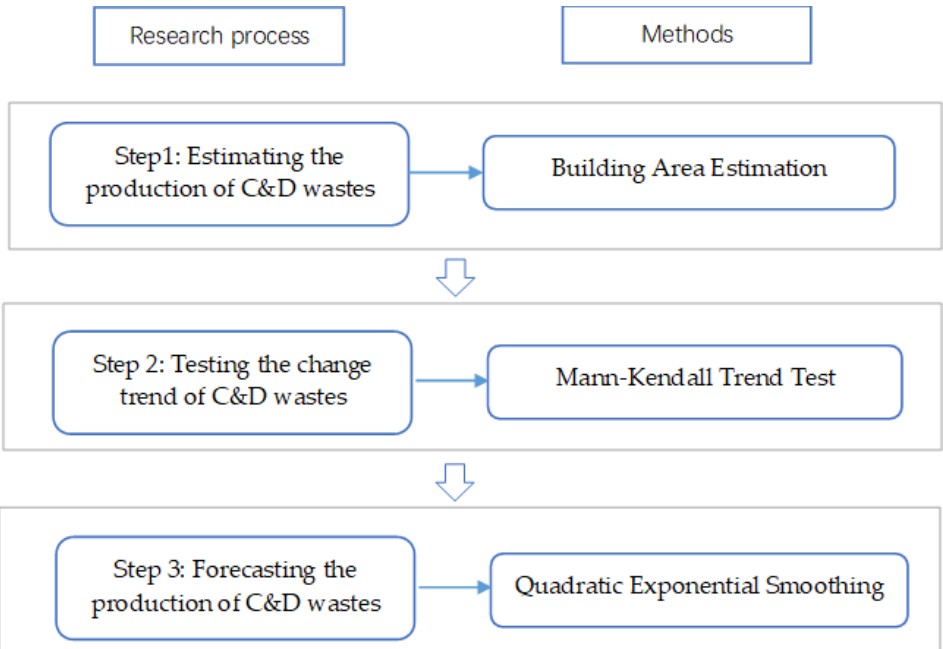

**Figure 1.** Research route map of construction and demolition (C&D) waste in Shandong Province.

### 2.1. Building Area Estimation Method

The building area estimation method was adopted to estimate the amount of C&D waste generated in Shandong Province. Firstly, according to Shandong Statistical Yearbook, the data of housing construction area and completed area in Shandong Province from 2000 to 2017 were obtained. Then, assuming that the annual demolition area was 10% of the construction area of the year, the C&D waste output per unit construction area was 550 tons, and the construction waste output per unit area was 13,000 tons, the data were processed to estimate the annual generation of C&D waste [35,36].

$$W = W_c + W_d \tag{1}$$

$$W_c = S_c \times q \tag{2}$$

$$W_d = S_d \times w \tag{3}$$

where $W$ means the annual production of C&D waste, $W_c$ means the annual production of construction waste, $W_d$ means the annual production of demolition waste. $S_c$ is the annual construction area, $S_d$ is the annual demolition area, according to the 10% of the construction area of the new building in the same year. $q$ refers to the amount of C&D waste generated per unit construction area (550 tons) [35], and $w$ refers to the amount of C&D waste generated per unit demolition area (approximately 13,000 tons) [37].

The data of the building construction and completion areas were obtained from the Statistical Yearbook of Shandong Province.

### 2.2. Mann–Kendall Trend Test

For time series X, the statistics $S$ of Mann–Kendall trend test [38,39] are as follows

$$S = \sum_{i=1}^{n-1} \sum_{j=i+1}^{n} sgn(x_j - x_i) \tag{4}$$

where $x_i$, $x_j$ are the data values of the time series, $n$ is the number of data points, and $sgn$ is a symbolic function defined as

$$sgn(x_j - x_i) = \begin{cases} 1, & if \ x_j - x_i > 0 \\ 0, & if \ x_j - x_i = 0 \\ -1, & if \ x_j - x_i < 0 \end{cases} \tag{5}$$

The variance is calculated as follows

$$Var(S) = \frac{n(n-1)(2n+5) - \sum_{p-1}^{g} t_p(t_p - 1)(2t_p + 5)}{18} \tag{6}$$

where $n$ is the number of data point, $g$ is the number of tied groups, and $t_p$ is the number of observations in group $p$.

The standardized statistics are calculated as follows

$$Z_c = \begin{cases} \frac{S-1}{\sqrt{Var(S)}}, & S > 0 \\ 0, & S = 0 \\ \frac{S+1}{\sqrt{Var(S)}}, & S < 0 \end{cases} \tag{7}$$

If $Z_c > 0$, the sequence is an upward trend, and if $Z_c < 0$, the sequence is a downward trend. The absolute value of Z is more than 2.32, 1.64 and 1.28, which means passing the significance test with confidence of 99%, 95% and 90% respectively.

The advantage of Mann–Kendall test is that the sample does not need to follow a certain distribution and can be easily calculated because it is not interfered by few abnormal values [39,40].

This approach, which was performed using MATLAB in this study, will be used to investigate the trend of building waste production [41,42].

### 2.3. Exponential Smoothing Prediction Method

Exponential smoothing prediction method is a forecasting method based on the actual and forecasted numbers of the current period of a certain index that introduces a simplified weighting factor (i.e., smoothing coefficient) to obtain the average number. This technique can eliminate the accidental changes in the time series, reflect the trend of the latest data, and accurately grasp the short-term predictions [38]. This prediction method is divided into three types: one-time, quadratic, and cubic exponential smoothing prediction models, which are applied to data that randomly fluctuates around a certain level, display a continuous linear trend, and follow a continuous trend of curve changes, respectively.

The trend of the C&D waste production in Shandong Province from 2000 to 2017 demonstrates a continuous linear change. The quadratic exponential smoothing method uses the weighted average of historical data as the prediction result at the future moment. Considering the changes of linear parameters of time series in different periods, the predicted values fit well with the original time series. It has the advantages of simple calculation, less sample requirements, strong adaptability, and stable results. Therefore, the quadratic exponential smoothing model was selected for the prediction of the waste production.

The accuracy of the model is the basis for the next step of the prediction. The methods for evaluating the model prediction accuracy include relative error, correlation, and posterior difference test methods. In this study, the relative error and posterior difference methods were used to evaluate the prediction model. In the former, the model prediction accuracy is considered high if the relative error is less than 10%. Two parameters were used when applying the latter to determine the model accuracy, namely, mean square error ratio C and small error probability P [6]. The accuracy level of the model is presented in Table 1.

**Table 1.** Model prediction accuracy level.

| Model Accuracy Level | Mean Square Error Ratio C | Small Error Probability P |
|---|---|---|
| Excellent | <0.35 | 0.95–1 |
| Qualified | 0.35–0.5 | 0.8–0.95 |
| Reluctantly | 0.5–0.65 | 0.7–0.8 |
| Unqualified | >0.65 | <0.7 |

## 3. Results and Discussion

### 3.1. Analysis of C&D Waste Production

According to the statistical yearbook of each province, the annual construction area in Shandong Province is in the forefront of China every year but the utilization rate of the C&D waste resources is relatively low. In 2015, Shandong Province's C&D waste output was about 100 million tons, and the resource utilization rate was only 8%. Therefore, the province was selected as the case study to analyze the waste production process and provide data support for the utilization of C&D waste resources.

Along with the data of building construction and completed areas obtained from the Statistical Yearbook of Shandong Province, the building area method (Equations (1)–(3)) was used to estimate the production of building waste in Shandong Province for the past 18 years (2000–2017). The results are listed in Table 2. The current situation of the C&D waste generation indicates that the annual amount of C&D waste has exceeded 110 million tons. Without considering the utilization of resources, the amount of accumulated C&D waste for the past 18 years has reached 1.1 billion tons.

**Table 2.** Estimated annual construction waste production in Shandong Province in 2000–2017. Unit: 1000 tons.

| Year | 2000 | 2001 | 2002 | 2003 | 2004 | 2005 | 2006 | 2007 | 2008 | 2009 | 2010 | 2011 | 2012 | 2013 | 2014 | 2015 | 2016 | 2017 |
|---|---|---|---|---|---|---|---|---|---|---|---|---|---|---|---|---|---|---|
| Total | 18,143.6 | 21,085.0 | 24,175.0 | 29,355.6 | 30,260.7 | 39,620.6 | 45,538.2 | 50,271.4 | 53,568.2 | 59,785.5 | 68,826.2 | 76,259.1 | 85,817.5 | 97,022.4 | 105,729.7 | 103,218.3 | 106,764.4 | 113,372.0 |
| Jinan | 2454.8 | 2849.5 | 2966.4 | 3932.2 | 3244.8 | 5057.6 | 5154.6 | 5820.9 | 6094.4 | 6510.7 | 6768.8 | 8222.7 | 9423.7 | 11,111.8 | 12,912.4 | 14,367.4 | 14,644.3 | 16,025.7 |
| Qingdao | 2659.6 | 3291.5 | 4130.0 | 4648.0 | 6260.8 | 5847.4 | 6561.2 | 8130.7 | 8333.8 | 9066.7 | 10,250.8 | 10,208.1 | 11,881.2 | 13,299.1 | 15,392.8 | 15,732.7 | 18,416.0 | 19,455.9 |
| Zibo | 1233.0 | 1639.7 | 1964.8 | 2583.5 | 1497.2 | 3723.3 | 5350.8 | 5271.4 | 4948.2 | 5583.7 | 6822.2 | 8071.7 | 9608.2 | 11,105.1 | 11,573.1 | 10,319.5 | 10,166.7 | 10,841.7 |
| Zaozhuang | 739.2 | 760.1 | 829.0 | 959.1 | 846.4 | 1458.6 | 1660.5 | 1997.4 | 2038.4 | 2283.5 | 2498.0 | 2846.6 | 3236.8 | 4023.5 | 3815.0 | 3705.8 | 3503.2 | 4086.7 |
| Dongying | 410.6 | 509.3 | 516.2 | 625.6 | 1590.0 | 960.1 | 1271.4 | 1245.7 | 1164.0 | 1262.1 | 1250.3 | 1299.3 | 1360.9 | 1505.2 | 1473.4 | 1292.5 | 1230.2 | 1058.5 |
| Yantai | 1585.0 | 2035.0 | 2286.0 | 2801.9 | 3086.6 | 3691.8 | 4119.7 | 4541.8 | 4711.0 | 4947.4 | 6109.5 | 6018.7 | 6112.9 | 6490.6 | 6124.8 | 5663.8 | 5857.0 | 6113.8 |
| Weifang | 1482.9 | 1829.1 | 2048.1 | 2520.6 | 2194.4 | 3425.0 | 4191.1 | 4624.0 | 5637.0 | 6970.8 | 8046.3 | 8812.1 | 9561.5 | 10,680.1 | 10,992.3 | 10,629.5 | 10,117.0 | 10,861.5 |
| Jining | 1027.0 | 963.9 | 1329.5 | 1520.2 | 1571.0 | 2174.4 | 2014.5 | 2415.7 | 2679.5 | 3089.0 | 4059.5 | 4278.6 | 5242.3 | 6675.8 | 7785.1 | 7615.5 | 7754.9 | 6959.7 |
| Taian | 1866.7 | 2025.7 | 2263.1 | 2149.1 | 1177.3 | 3067.1 | 3421.5 | 3715.0 | 4368.6 | 4842.7 | 5393.2 | 5736.9 | 5889.4 | 6005.1 | 5701.7 | 518.39 | 4624.7 | 4365.2 |
| Weihai | 803.6 | 935.7 | 1176.3 | 1228.3 | 2497.6 | 1845.3 | 2103.9 | 2542.9 | 2757.1 | 2934.7 | 3232.4 | 3814.4 | 3721.8 | 3887.1 | 3806.3 | 3588.9 | 3803.0 | 3705.8 |
| Rizhao | 304.0 | 417.4 | 470.8 | 565.0 | 722.7 | 1009.7 | 989.2 | 1082.7 | 1077.9 | 1233.5 | 1099.2 | 1433.1 | 1833.0 | 1986.1 | 2207.2 | 2247.8 | 2838.0 | 2860.8 |
| Laiwu | 269.1 | 279.8 | 351.9 | 448.9 | 232.4 | 539.2 | 765.9 | 808.0 | 709.0 | 708.2 | 674.5 | 720.9 | 1613.9 | 820.4 | 765.4 | 741.0 | 694.9 | 714.0 |
| Linyi | 1262.5 | 1377.4 | 1547.4 | 1767.4 | 137.54 | 2458.4 | 2852.7 | 3032.4 | 3446.4 | 3844.0 | 4993.9 | 5931.4 | 6725.5 | 8686.2 | 10,609.7 | 10,988.6 | 11,316.2 | 13,026.2 |
| Dezhou | 496.0 | 604.1 | 530.7 | 717.4 | 1076.1 | 1028.1 | 1053.0 | 1140.8 | 1390.0 | 1764.0 | 1958.5 | 2206.3 | 2382.8 | 2691.5 | 2956.9 | 2731.3 | 3016.5 | 3539.3 |
| Liaocheng | 552.0 | 599.3 | 747.3 | 1189.3 | 925.5 | 1121.5 | 1107.7 | 1159.5 | 116.17 | 1610.8 | 2157.1 | 2627.4 | 2958.9 | 3607.9 | 4882.4 | 3658.4 | 3788.3 | 4188.6 |
| Binzhou | 627.0 | 599.6 | 650.8 | 807.7 | 958.2 | 971.5 | 1362.6 | 1190.5 | 1270.9 | 1180.5 | 1363.0 | 1812.2 | 1968.5 | 1853.0 | 1955.6 | 1849.0 | 1873.1 | 1979.5 |
| Heze | 370.5 | 367.7 | 367.0 | 891.4 | 989.6 | 1241.6 | 1558.0 | 1551.9 | 1780.3 | 1953.4 | 2149.1 | 2218.7 | 2296.3 | 2594.0 | 2775.6 | 2902.7 | 3120.4 | 3588.9 |

### 3.1.1. Trend Analysis

The overall C&D waste production in Shandong Province from 2000–2017 displayed an increasing trend. The annual production increased from 18.1436 million tons in 2000 to 113.372 million tons in 2017, with an average annual growth rate of 11.38%. The Mann–Kendall trend test method (Equations (4)–(7)) was used to test the variation characteristics of C&D waste output. The results are shown in Table 3 ($Z_c$ = 15.24, greater than 2.32), indicating that the growth trend is very significant [43,44]. In addition, due to the different degrees of development of the social economy and urbanization in Shandong Province during different periods, the growth of the building waste production may also be different [25]. Therefore, this study further analyzed the change in building waste production every five years. Table 4 shows that the change in waste production displays an increasing trend from 18.53 million tons between 2001 and 2005 to 26.9592 million tons between 2011 and 2015.

The results of the Mann–Kendall trend test for the estimated building waste production in each city shown in Table 3 indicate that the annual building waste production in the 17 cities of Shandong Province exhibited a significant growth trend from 2000–2017, with an average annual growth rate of 5.1–14.7%.

**Table 3.** Results of the Mann–Kendall trend test for the variation characteristics of C&D waste production estimates.

| City | $Z_c$ | Trend | *p*-Value |
|---|---|---|---|
| Shandong Province | 15.24 | Rise significantly | <0.01 |
| Jinan | 5.27 | Rise significantly | <0.01 |
| Qingdao | 5.18 | Rise significantly | <0.01 |
| Zibo | 4.64 | Rise significantly | <0.01 |
| Zaozhuang | 5.00 | Rise significantly | <0.01 |
| Dongying | 3.38 | Rise significantly | <0.01 |
| Yantai | 4.73 | Rise significantly | <0.01 |
| Weifang | 5.09 | Rise significantly | <0.01 |
| Jining | 5.09 | Rise significantly | <0.01 |
| Taian | 4.19 | Rise significantly | <0.01 |
| Weihai | 4.55 | Rise significantly | <0.01 |
| Rizhao | 5.09 | Rise significantly | <0.01 |
| Laiwu | 3.11 | Rise significantly | <0.01 |
| Linyi | 5.09 | Rise significantly | <0.01 |
| Dezhou | 5.00 | Rise significantly | <0.01 |
| Liaocheng | 4.73 | Rise significantly | <0.01 |
| Binzhou | 4.37 | Rise significantly | <0.01 |
| Heze | 5.00 | Rise significantly | <0.01 |

Note: At a significance level of 0.01, if *p*-value < 0.01, it means that the sequence change trend is significant. *p*-value is the probability of Mann-Kendall statistic *S*.

**Table 4.** Changes in the C&D waste production in Shandong Province every five years from 2000–2017.

| Different Periods | 2001–2005 | 2006–2010 | 2011–2015 |
|---|---|---|---|
| Change of construction and demolition waste production (1000 tons) | (+)18,535.6 | (+)23,288.0 | (+)26,959.2 |

### 3.1.2. Spatial Distribution Characteristics Analysis

The spatial distribution of C&D waste in Shandong Province shows that cities with great quantity amounts of C&D waste are mainly concentrated in the areas with good economic development and high urbanization level in the central and eastern regions. Qingdao has the highest annual C&D waste production, exceeding 19 million tons; followed by Jinan, which produces more than 16 million tons; Zibo, Weifang, and Linyi with 10–15 million tons; Yantai and Jining, which produce 5–10 million tons; Tai'an, Zaozhuang, Dongying, Weihai, Rizhao, Dezhou, Liaocheng, Binzhou, and Heze are cities with 1–5 million tons; and lastly Laiwu City with less than 1 million tons.

Linyi has the highest average annual growth rate of construction waste (14.7%); followed by Heze, Rizhao and Zibo (between 13.6% and 14.3%); Liaocheng, Weifang, Qingdao, Dezhou, Jining, Jinan and Zaozhuang (between 10.6% and 12.7%), and the remaining six cities (under 10%). The average annual growth rates of 11 out of the 17 cities exceed the average growth rate. As the most populated and largest city in the province, Linyi has shown remarkable results in economic development and urban construction in the recent years. The amount of C&D waste in the city rapidly increased and its current waste production ranks third among the 17 cities. The development strategy of the western economic uplift zone in Shandong Province facilitated the development of the social economy of Jining. The amount of C&D waste generated by project constructions in the city increases at an average growth rate of 11.9%. Although Heze and Rizhao generate small amounts of C&D waste, the average growth rate of the two cities is the second and third respectively. With the urban infrastructure construction in the two cities, the C&D waste generation is likely to increase in the future. Although the average growth rate of C&D waste generation in Qingdao and Jinan is at a medium level, it has a large base. Considering that Qingdao is a sponge city and a comprehensive pipeline gallery as a national "double pilot" city and Jinan is a national pilot city for sponge cities, both cities are currently accelerating their rail transit construction. Therefore, the production of C&D waste in these two cities will remain high in the future.

### 3.2. Prediction of C&D Waste Production

On the basis of the estimated amount of C&D waste of Shandong Province from 2006 to 2017, the quadratic exponential smoothing model was used to predict the amount of C&D waste. The comparison between the predicted and estimated values is presented in Table 5. The results show that the predicted values of C&D waste production are close to the estimated ones, the average relative error is 9.21%, and the model prediction accuracy is high. After the calculation, the mean variance ratio is C = 0.09 < 0.35 and the small error probability P = 1 > 0.95. In conclusion, the accuracy level of the model is satisfactory and is suitable for the prediction of C&D waste production.

**Table 5.** Comparison between the predicted and estimated values of C&D waste production in Shandong Province from 2000–2017. Unit: 1000 tons.

| Year | Estimated Value | Predicted Value | Absolute Error | Relative Error% |
|------|-----------------|-----------------|----------------|-----------------|
| 2000 | 18,143.6 | - | - | - |
| 2001 | 21,085.0 | 23,357.5 | 2272.5 | 10.78 |
| 2002 | 24,175.0 | 22,828.7 | −1346.3 | −5.57 |
| 2003 | 29,355.6 | 24,490.9 | −4864.7 | −16.57 |
| 2004 | 30,260.7 | 31,038.1 | 777.4 | 2.57 |
| 2005 | 39,620.6 | 40,898.4 | 1277.8 | 3.23 |
| 2006 | 45,538.2 | 36,366.0 | −9172.2 | −20.14 |
| 2007 | 50,271.4 | 57,655.4 | 7384.0 | 14.69 |
| 2008 | 53,568.2 | 62,909.7 | 9341.5 | 17.44 |
| 2009 | 59,785.5 | 64,902.9 | 5117.4 | 8.56 |
| 2010 | 68,826.2 | 64,646.2 | −4180.0 | −6.07 |
| 2011 | 76,259.1 | 74,446.1 | −1813.0 | −2.38 |
| 2012 | 85,817.5 | 89,913.8 | 4096.3 | 4.77 |
| 2013 | 97,022.4 | 97,161.5 | 139.1 | 0.14 |
| 2014 | 105,729.7 | 110,018.1 | 4288.4 | 4.06 |
| 2015 | 103,218.3 | 125,410.6 | 22,192.3 | 21.5 |
| 2016 | 106,764.4 | 131,497.5 | 24,733.1 | 23.17 |
| 2017 | 113,372.0 | 108,665.6 | −4706.4 | −4.15 |

To grasp the output of C&D waste generated in the late period of the 13th and 14th five-year plans, the amount of C&D waste in Shandong Province from 2018 to 2025 was predicted using the proposed C&D waste prediction model. The prediction results are summarized in Table 6. The forecast

shows that the output of C&D waste in the province will continue to increase in the next six years and reach 141 million tons by 2025, accounting for an increase of 37.9 million tons from the amount in 2015. During the 13th five-year plan period, the total amount of C&D waste in Shandong Province will reach 570 million tons, which is in line with the current renovation of residential areas, demolition of illegal buildings, and construction of key city projects. The government and the public have gradually reached a consensus regarding the issue of C&D waste disposal, in which both sectors agree that simple and landfill disposal methods are unsustainable and the utilization of C&D waste resources is the inevitable choice to achieve sustainable economic development.

**Table 6.** Predicted values of C&D waste production in Shandong Province from 2018–2025. Unit: 1000 tons.

| Year | 2018 | 2019 | 2020 | 2021 | 2022 | 2023 | 2024 | 2025 |
|---|---|---|---|---|---|---|---|---|
| Predicted value | 114,123.3 | 117,972.5 | 121,821.7 | 125,670.9 | 129,520.0 | 133,369.2 | 137,218.4 | 141,067.6 |

The factors affecting the amount of C&D waste include not only the building area, but also the economic level, urbanization level, population, and other factors [45]. This study considered only the building area factor in the estimation of C&D waste; the influence of other factors is disregarded. The accuracy of the estimation results therefore requires further investigation. The research on the estimation methods for the amount of C&D waste generated must be strengthened and the calculation standards for C&D waste quantity must be studied and formulated in accordance to the actual situation of the different regions. The quadratic exponential smoothing method exerts a good effect on short-term prediction, but the accuracies in the mid- and long-term prediction are low. To further increase improve the prediction accuracy for short-term prediction, a combined weight method can be adopted, that is, two or more different single models are separately used for the prediction and the independent prediction results are weighted and averaged to obtain a final prediction result. For medium and long-term forecasting, scenario analysis methods can be utilized by setting different development scenarios on the basis of the analysis of factors affecting the C&D waste.

## 4. Conclusions and Policy Implications

Knowing the amount of existing C&D waste and accurately predicting the waste amount in the next few years are the prerequisites for effective C&D waste management. In this study, the building area estimation method was used to estimate the generation of C&D waste in Shandong Province, and the quadratic exponential smoothing method was used to predict the future waste output.

The estimation result shows that the amount of C&D waste in Shandong Province is 110 million tons, with an annual growth rate of 11.38%. In addition, the average growth rate of the 17 cities ranges between 11% and 52%. Cities with relatively large amount of C&D waste are mainly concentrated in areas with good economic conditions and high urbanization rates in the central and eastern regions. Qingdao, Jinan, and Linyi are the top three cities with the highest production of C&D waste. Nine cities have an average annual growth rate that exceeds the average growth rate; Linyi has the highest growth rate, which exceeds 50%.

The prediction indicates that the amount of C&D waste in Shandong Province will continue to increases and reach 141 million tons by 2025. During the 13th five-year plan period, the total amount of C&D waste in the province has exceeded 570 million tons. Therefore, the C&D waste treatment and resource utilization should be strengthened in terms of formulation plans and standards, pilot demonstrations, technological innovation, and policy support.

The comprehensive utilization rate of C&D waste in Shandong Province is relatively low and the corresponding treatment and resource utilization will continue to increase. Hence, the following suggestions are proposed to promote the C&D waste treatment and resource utilization.

First, several indicators, such as construction waste production and resource utilization, should be included in the statistical category of the statistical department. The specific situation of construction

waste in various places should also be clarified to provide a basis for the related planning and labeling of construction waste resource utilization.

Second, related plans and standards should be implemented. The specific development plan for the utilization of C&D waste resources in Shandong Province should be formulated to guide C&D waste treatment and resource utilization works. The technical guidelines for C&D waste resource utilization, as well as for disposal plants, should be established to standardize the C&D waste collection, transportation, utilization, and disposal plant construction. Moreover, a joint control and co-management mechanism should be developed, the law enforcement guarantee system should be improved, and entire process license control should be imposed to construct an "entire process license management, comprehensive resource utilization, unified, and balanced consumption" of the construction waste collection, transportation, and disposal systems.

Third, the number of pilot demonstration cities should be increased. Tai'an and Linyi have been identified as demonstration cities for waste resource construction utilization in 2017. Additional demonstration cities should be certified and supported with increased financial support. In addition, demonstration parks and projects should be constructed, application of renewable products of construction waste should be encouraged, and the development of the C&D waste resource industry should be promoted.

Fourth, the research on technology integration and innovation should be strengthened. At present, most of the C&D waste resource utilization technologies and equipment are imported. Therefore, the research and development of key technologies, such as construction waste treatment and renewable product technologies, and the advanced applicable production equipment should be improved to continuously improve the technical level of the utilization of C&D waste resources.

Lastly, related support policies should be enhanced. The C&D waste disposal fee system should be confirmed with reference to the municipal solid waste disposal charging method. Preferential policies, such as tax reduction for C&D waste recycling products, land concessions for recycling plants, and product value-added tax refunds, should also be enacted. Qualified renewable products and recycled products in government-supported public welfare projects (e.g., public buildings, affordable housing projects, building energy conservation, green building demonstration projects, and urban public facilities) should be utilized. Furthermore, social capital should be introduced to participate in the utilization of C&D waste resources through public–private partnership or other applicable modes.

Buildings can be divided into commercial buildings, civil buildings, workshops, etc. according to their uses, and they can be divided into reinforced concrete structures, steel structures, wooden structures, etc. according to the type of structure. This study only estimated C&D waste output based on building area without considering the purpose and structure type of the building. The utilization of C&D waste needs to be classified and processed according to the composition. This study did not conduct an in-depth analysis of the composition of the C&D waste. These are the directions to be discussed and studied in the future.

**Author Contributions:** L.Q. and D.L. developed the idea and wrote the manuscript. X.Y., Q.W. revised the manuscript. Writing—review and editing, L.Q. and X.Y.; writing—original draft preparation, D.L.; formal analysis, X.Y.; supervision, Q.W. and Q.M. All authors have read and agreed to the published version of the manuscript.

**Funding:** This work was supported by National Key R&D Program of China (2019YFC1908100), Natural Science Foundation of China (71974116), Key R&D Project of Shandong Province (2018GSF122005, 2019RZE27013, 2019RKE27001) and Shandong Provincial Social Science Planning Research Project (20CGLJ13).

**Conflicts of Interest:** The authors declare no conflict of interest.

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
