# Peer review of "Generation and Prediction of Construction and Demolition Waste Using Exponential Smoothing Method: A Case Study of Shandong Province, China"

_sustainability, doi:10.3390/su12125094_

Round 1
Reviewer 1 Report
The paper entitled “Generation and Prediction of Construction and Demolition Wastes using Exponential Smoothing Method: A Case Study of Shandong Province, China” presents an interesting study that used Mann–Kendall trend test and exponential smoothing prediction to assess the generation and trend of construction and demolition waste in the Shandong Province. For this paper, some considerations should be taken into account:
- Abstract: Results could be presented as a percentage of increase or decrease in relation to the current generation of waste.
- Introduction: The differential of the work is not clear. What is the great difference of the method used in the present work in relation to the methods already used to determine the trend of generation of this type of waste? This method is generally applied in which type of study?
- Methods: line 116 to 124 - Description of the equation’s symbols could be in the paragraph form.;
- Methods: line 128 to 132 - More details about the Mann–Kendall test applied in the present work are needed;
- Methods: “The trend of the C&D waste production in Shandong Province from 2000 to 2017 demonstrates a continuous linear change. Therefore, the quadratic exponential smoothing model is selected for the prediction of the waste production” If the behavior is linear, why did you choose a quadratic exponential model? Explain.
- Results and Discussions: line 156 to 157 - “… the utilization rate of the C&D waste resources is relatively low”. How low is it? Specify;
- Results and Discussions: line 159 to 165 - This paragraph is confusing! Is the estimate in line with actual waste generation? How do the authors corroborate the results obtained in this analysis? Better discuss the results found together with reference that can validate the results found;
- Results and Discussions: line 170 - Replace “11.1372” by “113.372”;
- Results and Discussions: All results must be compared with references that present studies on the generation of waste.
Reviewer 2 Report
This article uses an exponential smoothing method to explore the generation and prediction of construction and demolition wastes in Shandong Province in China. The work is in the scope of the journal, however, redaction and structure of the paper should be improved as indicated below, especially, the methods and results should be clearer. The author must justify the following points:
Comment 1: A concise abstract is required to answer the following questions: What problem did you study and why is it important? What methods did you use? What were your main results? And what conclusions can you draw from your results? The author needs to reorganize the Abstract based on these basic questions.
Comment 2: The paper should be revised to highlight novelties. Please consider that this lack of novelty starts with the Abstract, Introduction, and Conclusion. The objectives and the scientific contribution of this work should be clarified more clearly in the Introduction Section.
Comment 3: The citation method should be unified for the whole work. Please, check this issue in the manuscript.
Comment 4: The author must identify the quadratic index smoothing method in the introduction and present its importance in the literature.
Comment 5: There is a need for an introduction text before subsection 2.1 to justify the further subsections and validate the reasons for using the Equations. Besides, I would suggest adding a Figure at the beginning of Section 2 to explain the proposed steps of applying the materials and methods to sort out the results. Such a Figure could make a better understanding of the applied methods to conduct or build up such analysis.
Comment 6: The application of the materials and methods in the Case Study are not outlined with necessary vigor. Justifying this point is highly important. The author needs to include sufficient methodological details in the paper, particularly Section 2, and elaborate on the produced results, in Section 3, from the proposed methods. There is a need to describe the case study in Section 2 before presenting the output results. This can be answered by using the statistical yearbook of Shandong Province as listed in lines 125 and 126. This step is highly important to justify the values presented in Table 5 and Table 6. Furthermore, it is highly important to present the way of applying Equation 2 and Equation 3 in the analysis to sort out the results. Subsection 3.1.1, including Table 3 and Table 4, is highlighting a major data that help to sort out the results of this work, however, references are missing herein.
Comment 7: In the Conclusion Section, the author needs to summarize the novelties, objectives and scientific contribution of this work, as well as highlighting the materials and methods proposed and applied in the study, before presenting and describing the case study and the output results. The last paragraph herein is to present the major limitations and the future recommendation. Hence, I would strongly suggest rewriting the Conclusion Section according to this flow of information in a text form to make the work citable by other researchers. The reader might read this Section of the study only and must understand the whole manuscript.
Reviewer 3 Report
The study appears interesting and more about this topic should be done and implemented by administration and policy system to preview and estimate the quantity of CDW produced, in order to establish future policy and devices to adequate face this big problem.
I suppose this study can be extendend to all provinces in China, one of the most important worldwide "waste producers" so the future residues policy in the country can take into account these forecasts.
Round 2
Reviewer 2 Report
The authors have answered almost all my comments. Some other comments are left, as follows:
Comment 1: How Research Process is connected to Methods in Figure 1. Please show up the relation between these two lists.
Comment 2: In line 194, it says "the building area method (Eq. 1) was used to estimate..." Similarly, there is an need to justify the utilization of all other Equations presented in the previous section to prove the output results.
Comment 3: Table 3 and Table 4 need to be referenced, please.
